# Composition and Potential Function of Fecal Bacterial Microbiota from Six Bird Species

**Jose F. Garcia-Mazcorro** [1,*] 🆔**, Cecilia Alanis-Lopez** [2]**, Alicia G. Marroquin-Cardona** [3] **and Jorge R. Kawas** [4]

1  Research and Development, MNA de Mexico, San Nicolas de los Garza 66477, Mexico
2  Specialized Medical Center, Protection, Health and Animal Welfare, San Nicolas de los Garza 66450, Mexico; mvz.alanis.c@hotmail.com
3  Department of Physiology, Pharmacology and Toxicology, Faculty of Veterinary Medicine, Universidad Autonoma de Nuevo Leon (UANL), General Escobedo 66050, Mexico; aliciamarroquin@hotmail.com
4  Faculty of Agronomy, UANL, General Escobedo 66050, Mexico; jorge.kawas@mnademexico.com
*  Correspondence: josegarcia_mex@hotmail.com; Tel.: +52-81-8850-5204

**Simple Summary:** The digestive tract contains millions of microorganisms that are important for health and disease. Several bird species are commonly kept as pets, but little is known about the microorganisms present in their digestive tract. In this work, we present the most comprehensive survey of fecal microorganisms from pet birds to date. The results show evidence to suggest that (1) each bird species present a distinctive bacterial composition in feces, and (2) that this microbiota is associated with unique potential functions (e.g., the ability to form biofilms). The findings are important to better understand the significance of microbes on the health of birds but may also be relevant in a context of diseases that are transmitted between animals and humans.

**Abstract:** Gut microbial communities play a fundamental role in health and disease, but little is known about the gut microbiota of pet bird species. This is important to better understand the impact of microbes on birds' health but may also be relevant in a context of zoonoses. Total genomic DNA samples from pooled fecal samples from 30 flocks (4–7 pet birds per flock) representing over 150 birds of six different species (two Passeriformes: Northern Mockingbird (*Mimus polyglottos*) and Zebra Finch (*Taeniopygia guttata*), and four Psittaciformes: Lovebird (*Agapornis*, different species), Cockatiel (*Nymphicus hollandicus*), Red-rumped Parrot (*Psephotus haematonotus*), and Rose-ringed Parakeet (*Psittacula krameri*) were used for 16S rRNA gene analysis. Several taxa were found to be different among the bird species (e.g., lowest median of Lactobacillus: 2.2% in Cockatiels; highest median of Lactobacillus: 79.4% in Lovebirds). Despite marked differences among individual pooled samples, each bird species harbored a unique fecal bacterial composition, based on the analysis of UniFrac distances. A predictive approach of metagenomic function and organism-level microbiome phenotypes revealed several differences among the bird species (e.g., a higher proportion of proteobacteria with the potential to form biofilms in samples from Northern Mockingbirds). The results provide a useful catalog of fecal microbes from pet birds and encourage more research on this unexplored topic.

**Keywords:** avian microbiome; pet birds; lactic acid bacteria; gut microbiota; zoonoses; probiotics

## 1. Introduction

Most tissues and organs of birds and other animals are permanently colonized by a complex set of bacteria and other microorganisms that play a fundamental role in animal biology and evolution [1]. In particular, the avian gastrointestinal tract (GI tract) is inhabited by millions of microorganisms (the gut microbiota) that are important to study because of their involvement in health, immunity, adaptation, and overall survival in the environment [2,3]. The composition, diversity, and function of the gut microbiota is

related to several hosts (e.g., gut volume and host taxonomy [4,5]) and environmental (e.g., diet [6,7]) factors.

Microorganisms can be studied using traditional culture techniques, but cultivable communities have little utility in contemporary microbial ecology partly due to our inability to grow them in vitro [8]. On the other hand, molecular methods (e.g., DNA sequencing of universal marker genes) do not depend on cultivable communities and thus offer a suitable alternative to study microbial life in a wide range of environments. The gut microbiota has been well investigated using these methods in humans, rodents, and other mammals, but less research has been performed on the avian gut microbiome. Molecular methods have been used to study the chicken GI microbiota mostly for commercial reasons [9,10] and a growing number of studies have also used these methods to investigate the gut microbiota in other birds, including wild species [11–17]. Feces are often chosen to study the gut microbiota because of ease of sampling but the fecal microbiota does not usually represent the microbiota in the distal GI tract of mammals. However, it has been shown that fecal matter in birds is a useful alternative to analyze large intestinal microbes [18,19]. This is interesting because avian feces contain both feces and urine, and the cloaca contains a mixed population of microbes from the digestive, reproductive and urinary systems [20], therefore it would be expected that feces in birds contain microbes not only from the gut but also from other organs. This thought is based on the presence of a measurable microbiota in urine from humans and other mammals [21,22].

The term "pet bird" refers to birds housed and bred for ornamental use and mostly includes members of Passeriformes and Psittaciformes [23]. Small flocks of several bird species from these groups are commonly kept in small cages as pets and the owners have frequent and prolonged contact with their fecal matter, but few studies have investigated the fecal microbiota in pet birds [11,13,24]. This topic is important for a better understanding of the impact of microbes on birds' health and the development of products aiming to improve health and nutrient digestibility (e.g., probiotics), but may also be relevant in the context of zoonoses [23,25]. The objective of this study was to characterize the fecal microbiota of pet birds using high throughput 16S rRNA gene sequencing and to use this information to predict the potential functions of the fecal microorganisms.

## 2. Methods

### 2.1. Fecal Sampling

This study involved collection of fecal samples only and was conducted in compliance with the current Mexican legislation for the use of animals in research (NOM-062-ZOO-1999). Pooled fecal samples were collected from a total of 30 different privately-owned flocks (Table 1) containing various numbers of birds (4–7 birds per flock) from six bird species (two Passeriformes and four Psittaciformes) using the same methodology used in our previous publication [13]. These species were selected because they are the most common in our area and the birds were kept outdoors in small cages with water and food available ad libitum. All samples were collected from Monterrey and its metropolitan area (Northern Mexico) at a single time point within the same period and none of the birds were receiving antibiotics or other medications. Briefly, we changed the tray on the bottom of each cage, cleaned it, placed new clean paper on it, and waited for the animals to defecate. This procedure usually took 20–40 min until we gathered enough fecal material to fill one 2-mL sterile plastic tube. The tubes were filled with small aliquots of all droppings available. Please note that this methodology was not designed to discard potential sources of contamination (e.g., birds walking on feces) but instead to provide a view of the microbiota in feces of pet birds that people have contact with. The tubes were placed on ice, transported to our laboratory, and stored at −20 °C until DNA extraction.

**Table 1.** Summary of characteristics from each fecal sample.

| Sample | Diet | Number of Birds (Female:Male) | Estimated Age for Females and Males |
|---|---|---|---|
| MP1 | Commercial feed * | 4 (undetermined) | 1 year |
| MP2 | Commercial feed | 5 (undetermined) | 6 months–1 year |
| MP3 | Commercial feed | 6 (undetermined) | 2–8 months |
| MP4 | Commercial feed | 4 (undetermined) | 2 years |
| MP5 | Commercial feed | 5 (undetermined) | 6 months–1 year |
| TG1 | Canary grass (*Phalaris canariensis*) | 6 (undetermined) | 8 months |
| TG2 | Mixture of seeds † | 6 (undetermined) | Undetermined |
| TG3 | Mixture of seeds | 5 (undetermined) | 3–9 months |
| TG4 | Mixture of seeds | 7 (undetermined) | 6 months–1 year |
| TG5 | Mixture of seeds | 6 (undetermined) | 6 months–1 year |
| AS1 | Mixture of seeds | 5 (2:3) | Undetermined |
| AS2 | Mixture of seeds | 5 (undetermined) | 6 months–1 year |
| AS3 | Mixture of sun flower and other seeds | 6 (undetermined) | 6 months–1 year |
| AS4 | Mixture of seeds | 4 (undetermined) | Undetermined |
| AS5 | Mixture of sun flower and other seeds | 5 (undetermined) | 4–5 months |
| PH1 | Breeding paste, croquette, foxtail millet (*Setaria italica*), fruit, mixture of seeds, wholemeal bread | 4 (2:2) | 2–4 years |
| PH2 | Breeding paste, croquette, foxtail millet (*Setaria italica*), fruit, mixture of seeds, wholemeal bread | 4 (2:2) | 3–5 years |
| PH3 | Breeding paste, croquette, foxtail millet (*Setaria italica*), fruit, mixture of seeds, wholemeal bread | 4 (2:2) | 3–5 years |
| PH4 | Fruit, legumes, mixture of sun flower and other seeds | 4 (undetermined) | 1 year |
| PH5 | Fruit, legumes, mixture of sun flower and other seeds | 4 (undetermined) | 5 months–1 year |
| NH1 | Breeding paste, croquette, foxtail millet (*Setaria italica*), fruit, mixture of seeds, wholemeal bread | 6 (3:3) | 3–4 years (females), 1–2 years (males) |
| NH2 | Breeding paste, croquette, foxtail millet (*Setaria italica*), fruit, mixture of seeds, wholemeal bread | 5 (2:5) | 2–5 years |
| NH3 | Mixture of sun flower and other seeds | 6 (undetermined) | 1 year |
| NH4 | Sun flower seeds | 6 (3:3) | 5–6 months |
| NH5 | Mixture of sun flower and other seeds | 6 (4:2) | 3 months–1 year |
| PK1 | Fruits, legumes, mixture of sun flower and other seeds | 4 (2:2) | 2–4 years |
| PK2 | Fruits, legumes, mixture of sun flower and other seeds | 4 (2:2) | 2–4 years (females), 1–5 years (males) |
| PK3 | Soft fruit, KAYTEE exact hand-feeding formula | 4 (undetermined) | 4–5 months |
| PK4 | Fruits, legumes, mixture of sun flower and other seeds | 4 (undetermined) | 1 year |
| PK5 | Fruits, legumes, mixture of sun flower and other seeds | 5 (undetermined) | 1–2 years |

All samples came from birds of different owners. Passeriformes: Northern Mockingbird (*Mimus polyglottos*, code MP) and Zebra Finch (*Taeniopygia guttata*, code TG). Psittaciformes: Lovebirds (*Agapornis*, code AS), Red-rumped Parrot (*Psephotus haematonotus*, code PH), Cockatiel (*Nymphicus hollandicus*, code NH), Rose-ringed Parakeet (*Psittacula krameri*, code PK). AS samples were obtained from different species of Agapornis (AS1: Yellow-collared Lovebird, *A. personatus*, AS2 and AS3: Fischer's Lovebird, *A. fischeri*, AS4 and AS5: Rosy-faced Lovebird, *A. roseicollis*). * Complete feed for laying hens based on ground cereals, cereal by-products and other ingredients with 16% minimum crude protein (Purina). † Mixture of seeds and other seeds refers to a commercial combination of birdseed (canary grass), linseed, red millet, white millet, oats, nyjer, and mustard seed.

### 2.2. DNA Extraction and Sequencing

Total genomic DNA was obtained from 100 mg of all pooled fecal samples (*n* = 30) using bead-beating followed by DNA purification using a commercial kit (Wizard Genomic DNA Purification Kit, Promega) [13]. Briefly, fecal samples were mixed with silica beads (0.1 mm) and lysis solution for 1 min in a high-speed homogenizer (FastPrep), followed by protein precipitation and DNA purification following the manufacturer's instructions. A negative control was included to assess potential contamination of laboratory reagents. Purified DNA samples were further processed at the Molecular Research LP (Shallowater, TX, USA). The semiconserved V4 region of the 16S rRNA gene was amplified using PCR with primers 515F (5′-GTGYCAGCMGCCGCGGTAA-3′) and 806R (5′-GGACTACNVGGGTWTCTAAT-3′). PCR products were checked in 2% agarose gel to confirm the success of amplification and the relative intensity of the bands. The amplicons were sequenced in a MiSeq instrument (Illumina, San Diego, CA, USA) at the Molecular Research LP as shown elsewhere [13].

### 2.3. Bioinformatics

Raw 16S sequence reads were processed and analyzed using default parameters in the open-source bioinformatics tool Quantitative Insights Into Microbial Ecology (QI-IME) [26]. The forward and reverse sequence reads were joined in QIIME using the join_paired_ends.py script. Demultiplex and quality filter was performed using the split_libraries_fastq.py script. Bacterial species (Operational Taxonomic Units, OTUs at 97% similarity) were selected based on the GreenGenes v.13.5 16S database using two approaches. First, using UCLUST as implemented in QIIME in the open-reference clustering algorithm described by Rideout et al. [27] for OTU description, alpha, and beta diversity. Second, using a closed OTU picking approach for further analysis in PICRUSt and BugBase predictions (see below). Possible chimeras were removed using ChimeraSlayer in QIIME after OTU assignments. Despite the potential relevance of rare taxa [28] and Cyanobacteria [29] in gut microbial ecology, we removed all singletons (OTUs that appear only once) and Cyanobacteria sequences before analysis in an effort to remove potential sequencing errors and plant contaminants, as suggested by others [2]. The dataset generated for this study can be found in the Sequence Read Archive from the NCBI (BioProject: PRJNA637115).

### 2.4. Differences among Bird Species

The linear discriminant analysis (LDA) effect size (LEfSe) [30] in the Galaxy platform of the Huttenhower Laboratory was used to find microorganisms that consistently explain the differences between microbial communities among the bird species. LEfSe first uses the non-parametric Kruskal–Wallis sum-rank test to detect taxa with significant differential abundance with respect to the class of interest (e.g., bird species), followed by the use of the Wilcoxon rank-sum test to investigate biological significance. As a last step, LEfSe uses LDA to estimate the effect size of each differentially abundant taxa. A threshold of 3.0 on the logarithmic LDA score was used for discriminative features because in our experience the default of 2 may provide too many significant taxa that may not biologically significant. The input file for LEfSe analysis contained the relative abundances of taxa with a hierarchical structure from the open approach, filtered by all unassigned taxa (at all taxonomic levels) and very low abundant taxa (i.e., phyla with less than 0.04% relative abundances). Bird species was used as class, bird order as subclass, and cage as subject. We also used Multivariate Analysis by Linear Models (MaAsLin) also from the Huttenhower Laboratory to find associations between metadata (e.g., bird species, type of diet) and microbial community abundance. In this analysis, we used four variables: inclusion of fruits in diet, inclusion of seeds in diet, as well as the bird's order and species, with a significance threshold (maximum false discovery rate) of 0.05, a minimum relative abundance of 0.0001, and a minimum prevalence of 0.01. The core microbiome (i.e., OTUs present in all samples) was calculated using the OTU table from the open approach for each bird species in QIIME and the results were used to calculate and draw Venn diagrams using an online tool from Ghent University (http://bioinformatics.psb.ugent.be/webtools/Venn/).

### 2.5. Diversity Analysis

Alpha diversity was estimated using the number of OTUs at 97% similarity, the Chao1 metric, and the Shannon and PD whole tree diversity indexes and compared with SAS University Edition using data from all 10 default iterations from the 30 samples ($n = 300$ total) using the MIXED procedure with "cage" as a random effect. Multiple comparisons were adjusted using the Tukey test. The unique fraction metric (UniFrac) [31] was calculated in QIIME and used as a measure of beta diversity. Both weighted and unweighted UniFrac distances were used because they can provide different insights into the factors that differentiate microbial communities [32]. Principal coordinate analyses (PCoA) were performed in PAST v.3.25 [33] using the UniFrac distances. The non-parametric permutational multivariate analysis of variance (PERMANOVA), the analysis of similarities (ANOSIM), and the adonis tests were used for the analysis of the strength and statistical significance of sample

groupings using the compare_categories.py script with the UniFrac distance matrices. In the case of statistical significance, we compared the UniFrac distances between sample groupings using the make_distance_comparison_plots.py in QIIME.

*2.6. Prediction of Metabolic Profiles and Organism-Level Microbiome Phenotypes*

Phylogenetic Investigation of Microbial Communities by Reconstruction of Unobserved States (PICRUSt) [34] was used to predict the metabolic profile based on the 16S reads using the OTU table from the closed OTU picking approach described above, based on the Kyoto Encyclopedia of Genes and Genomes (KEGG) [35]. STAMP [36] was used to compare PICRUSt features using a conservative cutoff of 0.001 (adjusted *p* value). We also used BugBase [37] to predict organism-level microbiome phenotypes using the OTU table from the closed OTU picking approach.

## 3. Results

This study produced 1,935,991 raw 16S rRNA gene sequences. A total of 1,750,960 good-quality 16S reads (90.4% of raw sequences, min: 18,602, max: 197,593 sequences per sample; average 298 nucleotides each, min: 200, max: 469) were available for analysis after splitting libraries and quality filtering. These sequences were assigned to 9815 OTUs at 97% similarity using the open approach. As explained in methods, we removed Cyanobacteria to discard potential plant contaminants but this taxon deserves more attention. For instance, Cyanobacteria accounted for 3.3% of all OTUs (10,150 OTUs were detected before removal of Cyanobacteria, compared to the 9815 final OTUs) and an average of 6.7% of all reads (min: 0%, max: 42.7%). Additionally, a separate analysis of Cyanobacteria revealed significant differences among the bird species (see "The case of Cyanobacteria" and Figure S1 in Supplementary Information for more about this).

*3.1. Fecal Bacterial Composition*

There was a high variation among individual pooled fecal samples even within the same species (Figure 1). The majority of all the microbiota was composed of five phyla: Firmicutes (average: 57.7%), Proteobacteria (16.6%), Tenericutes (13.8%), Actinobacteria (7.4%), and Bacteroidetes (2.6%), which together accounted for 98.1% of all taxa. Less than 2% of reads were found from very low abundant taxa (e.g., Verrucomicrobia, Spirochaetes) and reads that were not assigned to any phylum (Figure 1). Despite marked differences among the samples, each bird species appeared to harbor a specific microbial composition in feces (Figure 2). LEfSe confirmed this observation revealing a high number of bacterial groups that were significantly different in each bird species, thus highlighting the fact that each bird species harbored a distinctive bacterial profile in the feces (Figure 3 and Figure S2 in Supplementary Information). The analysis in MaAsLin revealed a total of 15 taxa that were significantly different (false discovery rate <0.05) between the bird species (8 taxa related to Proteobacteria from MP samples, 6 taxa related to Tenericutes and Lactobacillales from NH samples, and only 1 taxon related to Bifidobacteriaceae from TG samples). No other variable (e.g., diet components, bird's order) was significantly associated with microbial abundances according to MaAsLin analysis.

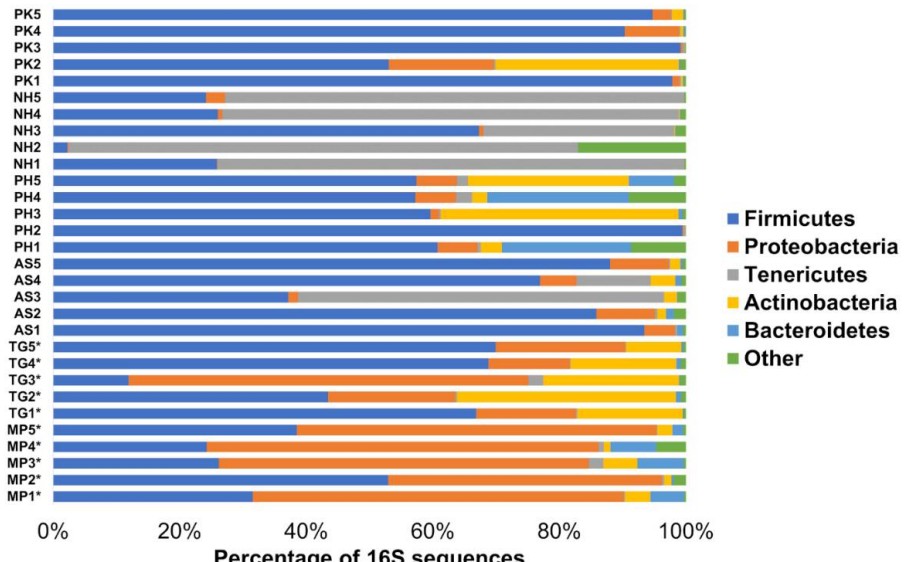

**Figure 1.** Bacterial composition (percentage of 16S sequences) of pooled fecal samples for each bird species at the phylum level. Most samples harbored high amounts of Firmicutes but Proteobacteria and Tenericutes dominated in samples from MP and NH, respectively. Samples from the two bird species of Passeriformes are highlighted to aid visualization (*) (see Table 1 for more information about each sample).

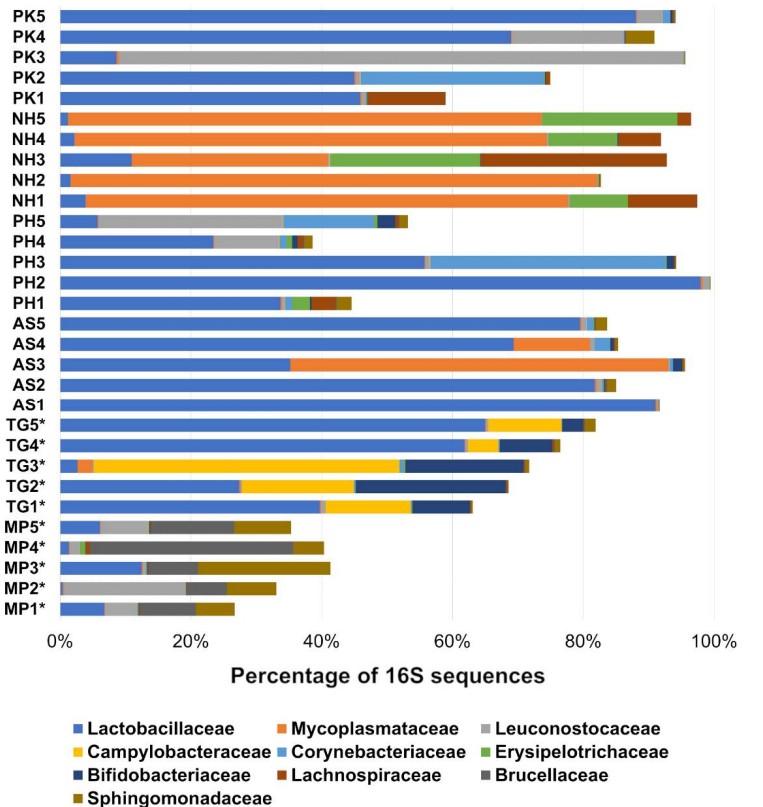

**Figure 2.** Bacterial composition (percentage of 16S sequences) of pooled fecal samples for each bird species at the family level. This figure only shows the 10 most abundant families calculated using data from all samples, therefore many other bacterial groups (the remaining to 100%) constitute each sample, especially in samples from MP. The differences within and between bird species are interesting with regards to the feed offered to the birds (e.g., MP samples were obtained from birds with a less varied diet). Samples from Passeriformes are highlighted to aid visualization (*) (see Table 1 for more information about each sample).

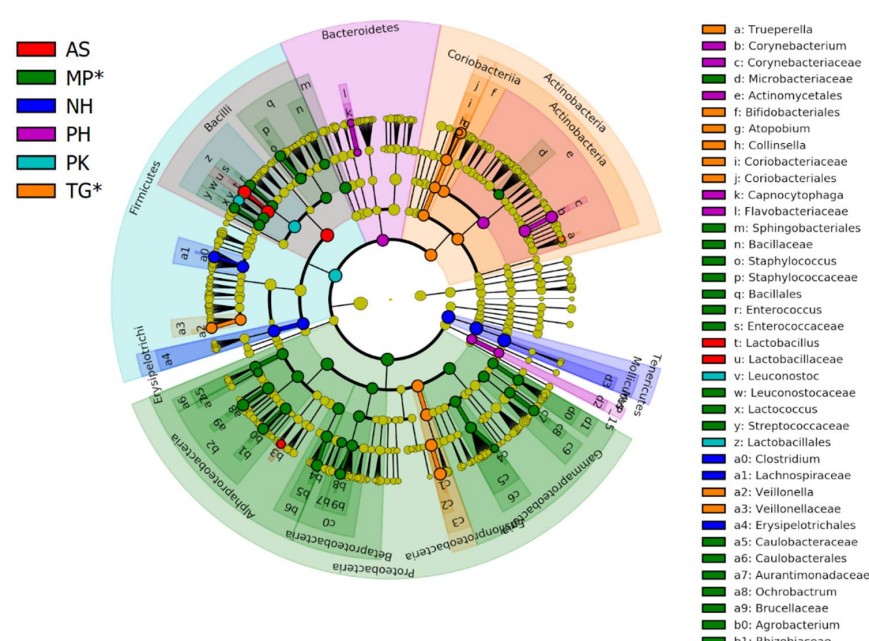

**Figure 3.** Cladogram showing the taxa that were significantly different among bird species according to LEfSe. The taxa are highlighted by small circles and by shading. LEfSe analysis generates taxa that are either significantly lower or significantly higher depending on the variable of interest. In this study, all taxa were higher (in other words, no taxa were found to be significantly lower in any bird species). Samples from Passeriformes are highlighted to aid visualization (*) (see Table 1 for more information about each sample).

In this study, we included samples from two species of Passeriformes, Northern Mockingbird (code MP), and Zebra Finch (code TG). Samples from MP were interesting because they harbored the most varied populations of microbes; for instance, the analysis of all data revealed that the first ten more abundant families comprised only 35% of the microbiota on average; in sharp contrast, the first ten more abundant families comprised 92% of all microbiota in samples from Cockatiels (Figure 2, see below). Additional analysis within each bird species confirmed the higher variety of microbes in MP samples (the first 10 more abundant families accounted for 64.2% of the reads, compared to >90% in all other bird species, min: 90% in TG samples, max: 99.2% in NH samples). Another characteristic of MP samples was the low abundance of Lactobacillaceae (median: 6.1%, min: 0.4%, max: 12.5%), which is interesting because most other samples, with the exception of Cockatiels (NH samples, see below), harbored higher abundances of this bacterial group. In contrast to MP samples, most samples from TG harbored higher amounts of Lactobacillaceae (median: 39.7%, min: 2.7%, max: 65.0%). Other peculiarities from TG samples include high amounts of Enterococcaceae (14.8%) in one sample (TG1) compared to the rest (0.1–3.1%), and high amounts of Enterobacteriaceae (12.7%) in one sample (TG3) compared to the rest (0.2–1.2%).

In this study we also included samples from four species of Psittaciformes. The fecal microbiota of Cockatiels (code NH) was dominated by very high amounts of *Mycoplasma* (median: 72.4%, min: 30%, max: 80.5%, Figure 2). Another interesting feature of NH samples was the low abundance (median: 2.2%) of Lactobacillaceae, which was also observed in the Passeriformes described above. This implies that the abundance of specific taxa may not predict the origin of the samples, in this case, the bird species. In contrast, samples from Lovebirds (code AS) were dominated by high amounts of *Lactobacillus* (median: 79.4%, min: 35.2%, max: 91.0%) and two samples contained high amounts of *Mycoplasma* (11.7% and 57.8%). Similar to AS samples, samples from *P. krameri* (code PK) also showed high amounts of *Lactobacillus* (median: 45.9%, min: 8.7%, max: 88.0%, Figure 2). Interestingly, two samples from AS had high amounts of the lactic-acid bacteria *Leuconostoc* (10.4% and 85.2%). Finally, samples from red-rumped parrot (code PH) also showed high

amounts of *Lactobacillus* (median: 33.7%, min: 5.5%, max: 97.4%), and similarly to PK samples, two samples from PH contained high amounts of *Leuconostoc* (4.5% and 23.7%, Figure 2).

### 3.2. Core Microbiome

The core microbiome is a concept that refers to a possible set of microbes that are common to all members of the same habitat, for example, the gut from a given animal species. The number of OTUs present in all 30 samples (i.e., shared OTUs) was only a minor proportion (0.3%) of all OTUs (32 of 9815) but the shared OTUs varied widely between the bird species (NH samples only shared 61 OTUs, PK: 91, AS: 116, PH: 120, MP: 205, TG: 226). These results are interesting because the two species of Passeriformes (MP and TG) showed the highest number of shared OTUs (205 for MP and 226 for TG) compared to the number of shared OTUs in the four species of Psittaciformes (61–120). Moreover, there was a linear relationship ($R^2 = 0.63$) between the average number of OTUs and the number of shared OTUs, which makes sense (the higher the number of OTUs the higher the number of shared OTUs). However, the two Passeriformes were clear outliers in this relationship and their removal produced a much stronger relationship ($R^2 = 0.98$ from the analysis of the four Psittaciformes). Overall, this implies that inter-species differences may not only comprise variations in the abundance or prevalence of taxa, or diversity metrics (e.g., the number of OTUs), but also in the numbers and types of microbes shared among individuals from the same species.

A total of 394 unique (i.e., not shared) OTUs were found among all samples. In contrast, 118 unique OTUs resulted from the analysis of all four species of Psittaciformes with only 34 shared OTUs (Figure 4). The inclusion of the two species of Passeriformes in this diagram did not change much the number of shared OTUs (32 and 34) but both showed a higher number of unique OTUs (95 for MP and 112 for TG) compared to the number of unique OTUs in each of the four species of Psittaciformes (6–29, Figure 4). In contrast, the 2 species of Passeriformes shared a total of 116 OTUs with 203 unique OTUs (Figure 4). Overall, these results indicate that the two species of Passeriformes were more similar to each other (higher numbers of shared OTUs) and more unique (higher numbers of unique OTUs) compared to the four species of Psittaciformes (lower numbers of shared and unique OTUs).

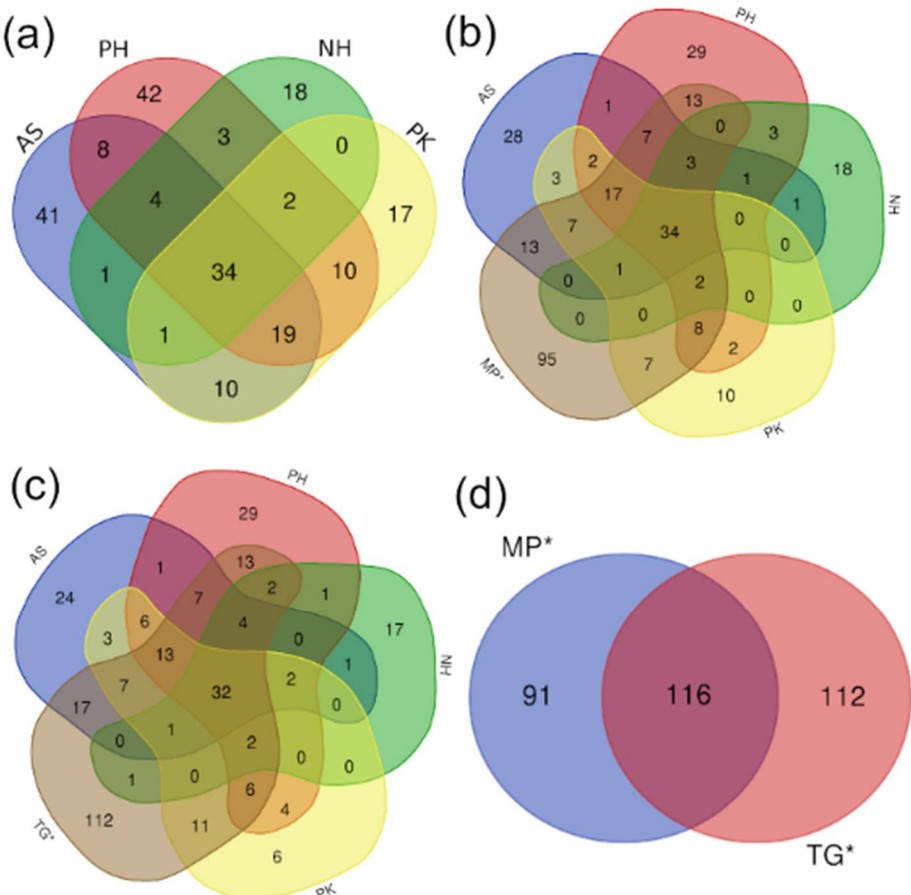

**Figure 4.** Venn diagrams using data from OTUs that were present in all samples from each bird species. (**a**) Diagram for all four species of Psittaciformes. (**b**) Diagram from all four species of Psittaciformes plus MP (Passeriformes). (**c**) Diagram from all four species of Psittaciformes plus TG (Passeriformes). (**d**) Diagram from the two species of Passeriformes (MP and TG). Samples from Passeriformes are highlighted to aid visualization (*) (see Table 1 for more information about each sample).

### 3.3. Diversity Analyses

The number of OTUs at 97% similarity (from the open approach) using a rarefaction depth of 18,000 sequences per sample (lowest number of sequences found in our samples) did not reach a plateau for 5 out of the 6 bird species (Figure 5), with samples from Cockatiels (NH) showing a very different pattern compared to all other species. This is in line with what we observed in the relative contribution of each taxa to the samples (Figure 2). The comparison of number of OTUs in SAS revealed a significant difference ($p = 0.03$) among bird species, but Tukey's adjusted multiple comparisons only revealed significance for the comparison between NH and MP ($p = 0.03$, higher in MP) and between NH and PH ($p = 0.04$, higher in PH). The comparison of Chao1 metrics ($p = 0.04$), PD whole tree ($p = 0.04$), and Shannon ($p = 0.002$) diversity indexes, also revealed significant differences among bird species, with MP and PH samples higher than NH ($p < 0.05$, see Figure S3 in Supplementary Information).

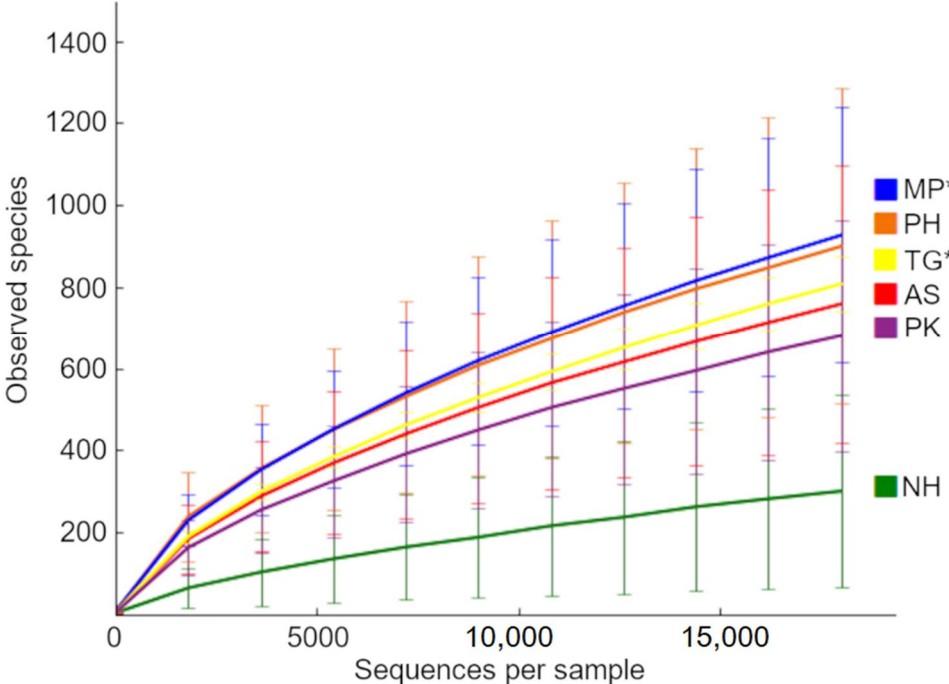

**Figure 5.** Rarefaction plots of OTUs at 97% similarity. Samples from Passeriformes are highlighted to aid visualization (*) (see Table 1 for more information about each sample). We repeated this analysis without rarefaction but even the bird species with the highest number of sequences (TG) still did not reach a plateau, thus implying that more sequences are needed to full describe the whole variety of OTUs.

Despite some overlap between samples, there was a significant clustering of samples using PCoA plots (Figure 6) and bootstrapped trees (see Figures S4 and S5 in Supplementary Information) accordingly to bird species. This clustering was confirmed using the PERMANOVA, ANOSIM (weighted R = 0.59, unweighted R = 0.41), and adonis tests using both weighted and unweighted UniFrac distances ($p$ = 0.001 for all tests). Additional comparisons between sample groupings revealed that these differences in UniFrac distances were mostly driven by NH samples (see Figure S6 in Supplementary Information). The clustering by bird type (Passeriformes and Psittaciformes) was also significant according to PERMANOVA ($p$ = 0.004 weighted, $p$ = 0.013 unweighted) and adonis tests (weighted $p$ = 0.002, unweighted $p$ = 0.022), but not ANOSIM (weighted $p$ = 0.067, R = 0.15; unweighted $p$ = 0.201, R = 0.07).

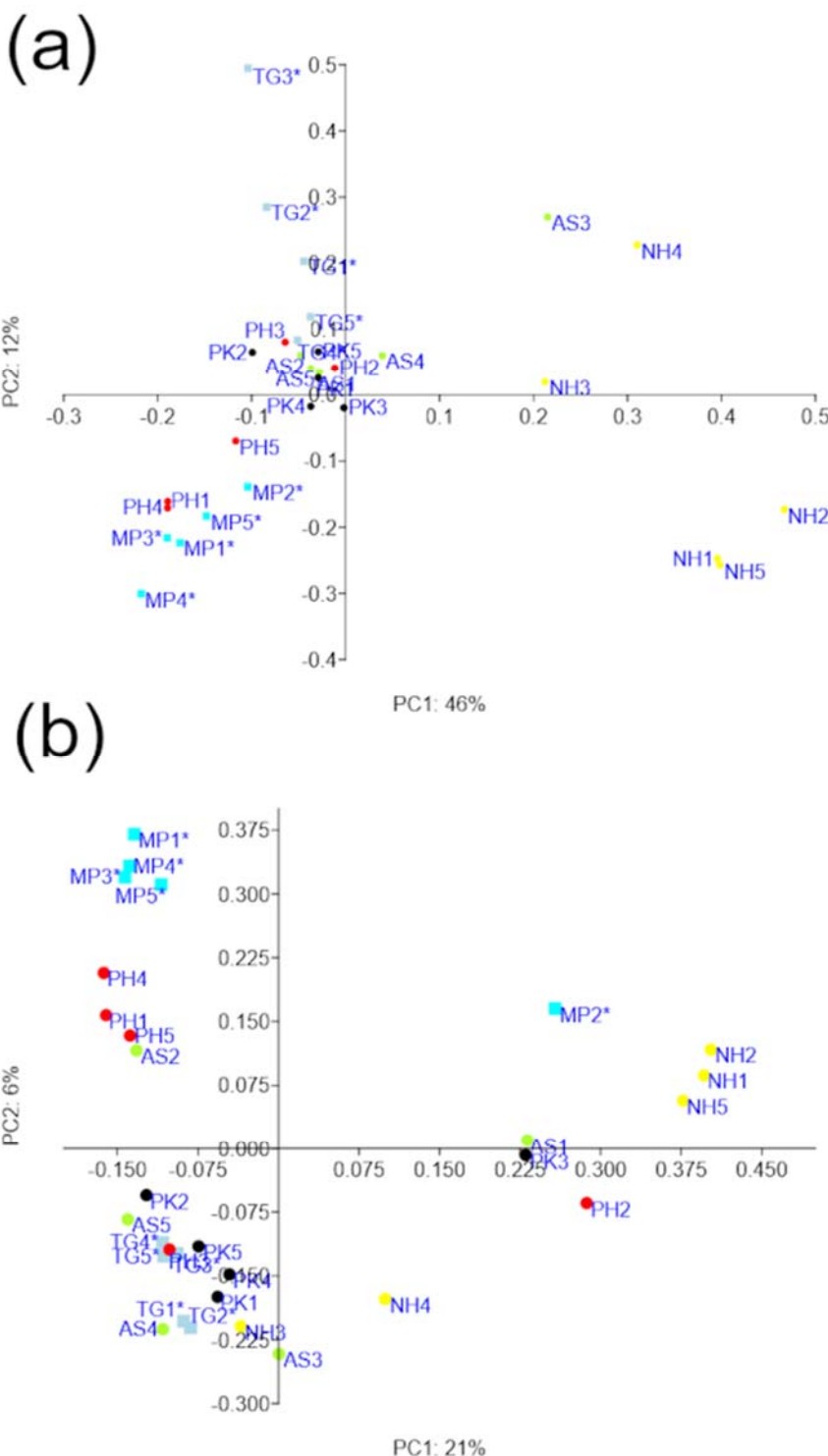

**Figure 6.** PCoA plots using weighted (**a**) and unweighted (**b**) UniFrac distances. PERMANOVA, ANOSIM, and adonis tests using both weighted and unweighted UniFrac distances revealed a significant clustering based on bird species (*p* = 0.001 for all tests) but additional comparisons showed that these differences were mostly driven by NH samples (see main text for more details). Please note that the use of other metrics (e.g., Bray–Curtis) may provide additional insights into the clustering of microbial communities. Samples from Passeriformes are highlighted to aid visualization (*) (see Table 1 for more information about each sample).

*3.4. Predicted Metabolic Profile*

The closed OTU picking approach yielded a lower number of OTUs (1828 vs. 9815 from the open approach) because this approach relies on an exact match against reference sequences. Using these OTUs, PICRUSt revealed a total of 52 features that were significantly different among bird species (Bonferroni adjusted $p < 0.001$, see Table S1 in Supplementary Information). Interestingly, samples from Northern Mockingbird showed the highest proportion for most features, perhaps because this bird species harbored the most variable bacterial composition. The use of BugBase revealed interesting data about the organism-level microbiome phenotypes associated with our samples. For example, aerobic Proteobacteria were only present in samples from MP and TG, anaerobic Actinobacteria were only present in samples from TG, and Proteobacteria with the potential to form biofilms was overrepresented in MP samples (see Figures S7–S15 in Supplementary Information).

**4. Discussion**

The gut microbiota is important for health and disease, but most research work on this subject has been performed in mammals compared to birds and other animal species. Several bird species are commonly kept in small flocks as pets throughout the world and fed diets consisting of one or few ingredients. While several papers have investigated the fecal microbiota of pet birds, to our knowledge this work represents the most comprehensive molecular survey of fecal microorganisms from pet birds living in their regular environment.

The numbers and types of microorganisms in any given environment (e.g., the avian gut) is a relevant subject, particularly in light of new research suggesting a relationship with the size (i.e., gut volume) of the ecosystem [5]. Almost 10 years ago it was suggested that our Planet harbor about 10,000 bacterial species [38] while others proposed 5.6 million OTUs as the lower bound of the microbial diversity on Earth [27]. More recent studies have found about 300 thousand unique sequences (also known as single-nucleotide resolution "sub-OTUs" or amplicon sequence variants) from multiple environments on Earth [39]. On the other hand, it has been proposed that the human gut microbiota contain 15,000 to 36,000 species [40] but more recent estimates suggest only 4930 species of bacteria in that environment [41]. The differences among these estimates are likely not derived from true differences in nature but from the methods used to catalog genetic sequences from marker genes, such as the 16S rRNA gene [42]. The number of OTUs in this study from the open approach (9815) contrasts with the 8751 OTUs in our previous study of three pet bird species [13], in which we used a similar methodology and the same sequencing instrument. These numbers also contrast with the 1828 OTUs detected using the closed OTU picking approach, because this approach relies on an exact match against reference sequences, and with the few hundreds (<500) from one meta-analysis of the avian microbiota that used clone-library and amplicon pyrosequencing data [2].

Birds are fascinating animals that differ widely in behavior, dietary preferences and patterns, flight capacity, and other traits that could potentially influence the presence, interactions, and functions of gut microorganisms. Interestingly, it has been proposed that birds may possess a wider range of microbiota composition compared to mammals, with the argument that there are more species of birds compared to mammals, adaptations to long-distance flights, and possible greater dependence on microbes to digest the food [15]. However, a direct comparison between birds and mammals may not be appropriate because of the wide differences within each group. Another potentially useful comparison involves wild vs. domestic birds, but a comparative analysis of the gut microbiota only revealed little differences between the two and those differences could be masked by the high number of wild birds included in the comparison (from Penguins to Hoatzins) [15]. Other studies have attempted to compare the gut microbiota between wild vs. captive parrots [43] and raptors [44] but whether these results are applicable to other bird species is unclear. In this regard, our results indicate that, with the exception of NH samples, the number and diversity of OTUs were similar among the pet birds studied, including domesticated wild

representatives such as Northern Mockingbirds, but that each bird species harbored a unique combination of bacterial taxa in feces. Overall, it is likely that the gut microbiota of wild birds is different compared to domesticated birds due to differences in diet and other environmental factors, but this comparison may be biased by inter-individual and inter-species variations, season of sampling, health status, and other factors, as shown here and in other studies [4,13]. Moreover, this comparison preferably should include other aspects besides just the composition of the microbiota, such as interactions between microbes [45], or ecological properties such as redundancy and resilience [46].

The results on specific taxa such as *Lactobacillus* shed light on the gut microbial ecosystem of birds. Despite the fact that the feces of birds also contain urine, it has been shown that feces are useful to reflect the microbiota in the distal gut of birds. *Lactobacillus* are members of the so-called lactic acid Bacteria and are characterized by the formation of lactic acid as the only or main end product of carbohydrate metabolism. In humans, this group has a low abundance (0.01%) but this varies widely depends on location and health status [47], and in mammals it has been suggested that most *Lactobacillus* in the intestinal tract are not true intestinal inhabitants but instead microbes coming from exogenous sources [48]. Unfortunately, we know less about this taxon in birds. Our previous study on pet birds showed that *Lactobacillus* was one of the most abundant taxon, especially in budgerigars and canaries [13], and in one study of 59 neotropical bird species, *Lactobacillus* was found in 100% of large intestinal samples [4]. However, thorough reviews have discussed this only superficially (with a statement that *Lactobacillus* are expected in the gizzard because they tolerate acidic environments [15], based on findings in the chicken GI tract [49]), or not at all [3]. Another study using Zebra Finch showed that members of Lactobacillaceae and Bifidobacteriaceae were not well established in the gut communities of hand-reared hatchlings compared to chicks raised by their biological or foster parents [16], which is in line with this current study on TG samples showing a predominance of *Lactobacillus*. More research is needed on this subject particularly in light of new research showing that *Lactobacillus* provokes a polarizing effect on the cecal microbiome of chickens, as revealed by simultaneous positive and negative interactions with other autochthonous taxa [45].

The results in this report may also be of interest to companies that manufacture and commercialize probiotics for pet birds and other animal species. In domestic poultry such as chickens, many studies have analyzed the effect of different probiotic formulations on health and productivity [50] but the efficacy of these products and relationship with the autochthonous gut microbiota remain controversial [45]. However, to our knowledge, there are only a few studies addressing the effect of probiotics on health in non-poultry but still there are products that are not well funded with the knowledge we currently have on gut microbial composition and metabolic activity. For example, there is a commercial product from a company in California, USA, containing a proprietary probiotic blend (180 million or $1.8 \times 10^8$ colony-forming units/g of five strains of *Lactobacillus* and one strain each of *Enterococcus*, *Streptococcus*, and *Bifidobacterium*) intended to be administered to all domesticated birds, including pet birds. The reasons why the company chose those bacterial strains are unknown to us, but it may be of interest for the reader to know there are probiotics products for cats, dogs, and humans containing a very similar bacterial composition [51,52]. Our results on *Lactobacillus* and other lactic acid bacteria with probiotic potential such as *Leuconostoc* [53], may prove helpful to develop probiotic formulations specifically for pet birds.

Most members of the gut microbiota (e.g., *Lactobacillus*) in birds and other animals are generally harmless but may also contain either pathogenic microbes or microbes that could become pathogenic when encountering a susceptible individual. In this study, samples from Cockatiels showed high abundance of *Mycoplasma* and this is interesting because all the birds were clinically healthy, thus suggesting that the *Mycoplasma* were commensals and part of the autochthonous microbiota. In contrast, *Mycoplasma* was only a very minor component (0.1–1%) of the fecal microbiota from Cockatiels in a similar diet in our previous paper [13], thus suggesting the existence of other factors affecting the composition of the

fecal microbiota among flocks of Cockatiels and other bird species. Additionally, two samples from Lovebirds (also Psittaciformes) also showed high amounts of *Mycoplasma* (11.7% and 57.8%), thus suggesting that this taxon may be part of the autochthonous fecal microbiota in individuals from different bird species. Although some species of *Mycoplasma* are well-known pathogens in birds [54], one *Mycoplasma* species (i.e., *M. gallisepticum*) has been detected in several species of wild birds with and without history of clinical signs [55] further supporting the idea of ubiquity of *Mycoplasma* among birds.

The analysis of shared OTUs revealed intriguing insights into the complexity of the avian microbiome. To look for similarities or differences within or between animal species, it is common practice to look at the relative abundance of specific taxa and/or diversity measurements. However, individuals may also be similar to one another in other aspects. For instance, this study showed that the two species of Passeriformes showed the highest number of shared OTUs (205 for MP and 226 for TG) compared to the number of shared OTUs in the four species of Psittaciformes (61–120). Overall, this implies that inter-species differences may not only comprise variations in the abundance or prevalence of taxa, or diversity metrics (e.g., the number of OTUs), but also in the numbers and types of microbes shared among individuals from the same species. A similar line of thought has been used in other studies looking at similarities in skin microbes between humans and dogs [56].

There are different ways to investigate the functions of the microbes in nature, for example using shotgun metagenome sequencing. This topic is important for various reasons, but it is particularly relevant because even identical 16S sequences can be found in Bacteria with highly divergent genomes and ecophysiologies [57]. Different tools have been developed to predict the function of microbial genes, including PICRUSt. Although these tools have been criticized because of the ambiguity of the predictions [58], others have discussed that the PICRUSt prediction framework is consistent with the known state of knowledge in avian microbiology [2]. It is interesting to note that PICRUSt predictions sometimes reveal little or no significant differences even in scenarios where a difference would be biologically feasible [59]. Therefore, the finding of differences in the predictions by PICRUSt, BugBase, and other tools deserve more attention.

There are several limitations of this current study. First, birds become anxious when separated from their flocks and this may impact physiological processes such as digestion, a phenomenon that can impact fecal consistency which has been shown to be the most influential factor affecting fecal microbiome variation in humans [60]. Therefore, we decided to use pooled fecal samples from all birds within each flock in order to investigate the fecal microbiota of pet birds that people more often have contact with, instead of the microbiota from artificially isolated individual birds. Second, diet and age are the most important factors influencing the gut microbiota, but in this study, we did not analyze the nutrient profile of each diet, did not investigate individual dietary preferences within a mixture of seeds, and the age of the birds was only an estimate. Third, we only analyzed samples at one time point but there is considerable over-time variation that needs to be taken into account. Future studies should consider these limitations to provide more details on the specific factors and conditions affecting fecal microbiota in pet birds.

## 5. Conclusions

In conclusion, our results provide a useful catalog of fecal bacteria in pet birds living in their regular environment. *Lactobacillus*, *Mycoplasma*, and other taxa, may be relevant groups to better understand the avian gut microbial ecosystem. The data suggest that each bird species carry a specific set of microbes in feces, and based on the available literature, it is likely that these microbes are representatives of the distal gut. Our data also suggests that these microbes are associated with unique potential functions, but this would have to be verified using other approaches besides predictions based on the 16S rRNA gene. The separate effect of age, diet, and other environmental factors, on the composition and function of the gut microbiota of pet birds, needs innovation in using isolated birds

displaying a similar physiology as when they are in a flock. The relevance of these findings to birds' health, potential zoonoses, and probiotic design is worth exploring.

**Supplementary Materials:** The following are available online at https://www.mdpi.com/2673-6004/2/1/3/s1, Figure S1: Box plots showing relative proportions of 16S reads from Cyanobacteria among all six bird species, Figure S2: Plot showing LEfSe results. Bars at the right show bacterial groups that were significantly higher and bars at the left show bacterial groups that were significantly lower, Figure S3: Rarefaction plots of Phylogenetic Diversity (PD) whole tree, Chao1, and Shannon diversity indexes. Samples from Passeriformes are highlighted (*), Figure S4: Bootstrapped tree using both weighted UniFrac distances. A total of 18,000 sequences were used in each jackknifed subset, Figure S5: Bootstrapped tree using unweighted UniFrac distances. A total of 18,000 sequences were used in each jackknifed subset, Figure S6: Box plots of weighted (a) and unweighted (b) UniFrac distances between sample groupings, Figure S7: OTU contributions from the four more abundant phyla and others for aerobic bacteria accordingly to BugBase analyses, Figure S8: OTU contributions from the four more abundant phyla and others for anaerobic bacte-ria accordingly to BugBase analyses, Figure S9: OTU contributions from the four more abundant phyla and others for facultative an-aerobic bacteria accordingly to BugBase analyses, Figure S10: OTU contributions from the four more abundant phyla and others for bacteria with potential to form biofilms accordingly to BugBase analyses, Figure S11: OTU contributions from the four more abundant phyla and others for gram-negative bacteria accordingly to BugBase analyses, Figure S12: OTU contributions from the four more abundant phyla and others for gram-positive bacteria accordingly to BugBase analyses, Figure S13: OTU contributions from the four more abundant phyla and others for bacteria with mobile elements accordingly to BugBase analyses, Figure S14: OTU contributions from the four more abundant phyla and others for potentially pathogenic bacteria accordingly to BugBase analyses, Figure S15: OTU contributions from the four more abundant phyla and others for stress tolerant bacteria accordingly to BugBase analyses, Table S1. Summary of those PICRUSt features with the lowest adjusted *p* values.

**Author Contributions:** Conceptualization, J.F.G.-M. and C.A.-L.; methodology, J.F.G.-M., C.A.-L., A.G.M.-C. and J.R.K.; software, J.F.G.-M. and J.R.K.; validation, C.A.-L., A.G.M.-C. and J.R.K.; formal analysis, J.F.G.-M., C.A.-L., A.G.M.-C. and J.R.K.; investigation, J.F.G.-M., C.A.-L., A.G.M.-C. and J.R.K.; resources, A.G.M.-C. and J.R.K.; data curation, J.F.G.-M. and C.A.-L.; writing—original draft preparation, J.F.G.-M. and C.A.-L.; writing—review and editing, J.F.G.-M., C.A.-L., A.G.M.-C. and J.R.K.; visualization, J.F.G.-M., C.A.-L., A.G.M.-C. and J.R.K.; supervision, C.A.-L.; project administration, J.F.G.-M.; funding acquisition, J.F.G.-M. All authors have read and agreed to the published version of the manuscript.

**Funding:** This study was partly funded by PRODEP (Secretary of Public Education, Mexico, grant number DSA/103.5/14/11021).

**Institutional Review Board Statement:** This study was conducted in compliance with the current Mexican legislation for the use of animals in research (NOM-062-ZOO-1999). Ethical review and approval were waived for this study, due to collection of fecal samples only (the birds were not handled in any way).

**Informed Consent Statement:** Not applicable.

**Data Availability Statement:** The data presented in this study are openly available in the Sequence Read Archive from the NCBI (BioProject: PRJNA637115).

**Acknowledgments:** We thank all the birds' owners for allowing the collection of fecal material from their pets.

**Conflicts of Interest:** The authors declare no conflict of interest.

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
