# Peer review of "Composition and Potential Function of Fecal Bacterial Microbiota from Six Bird Species"

_2673-6004, doi:10.3390/birds2010003_

Round 1

Reviewer 1 Report

Microbial  community  profiling  based  on  16S  ribosomal  rRNA  gene  sequencing has proven to be a reliable and reproducible representation  of  phylogenetic  and  taxonomic  categorization  of  bacteria. However, interpreting metagenomic data is a challenge since,  low  coverage  of  well-annotated  microbial  groups in available gene reference databases which makes it impossible to reliably assign all  phylotypes  and  functional  gene  identities  from  environmental  microbial  sequencing  libraries. Much lower number closed OTUs versus open OTUs  presented in reviewed  manuscript shows it clearly. The choice of the target region, algorithms and thresholds used can also lead to different clustering of OTUs. Thus, the conclusions drawn should be carefully. But all that things are most probably well known to Authors.    

Therefore, very important part of the manuscript should be the discussion chapter. In Introduction the Authors declare that their study has: (i) the cognitive goal -  'gut' avian microbiota and (ii) their function. In relation to the first part, generally, in my opinion the molecular, bioinformatics and statistic methods and the conclusions are correct.   

However, because it is well known, that both diet and individual age (the effect clearly visible in humans) have huge impact on gut microbiota of different animals including birds, I wonder, if Authors doing analyzes the data, didn't narrow down these parameters too much referring them only to the abundance of bacteria, which may lead to wrong conclusions in this aspect. Looking at the data obtained, it seems to be possible, that diet components influence microbiota of considered groups of birds. The group PK3 in comparison to other PK where much younger and on different diet (no legumes included which contain many bioactive compounds with different impact on microorganisms), also both PH4 and PH5 in comparison to other PH differed in age and diet, as well birds in MP4 group were much older than in other MP groups. Moreover, these factors don’t have to equally affect the individual groups of bacteria, but only some, not necessarily the most numerous. 

Therefore, I would suggest a more detailed and critical discussion on this aspect. In my opinion, it would be good to compare the obtained data with the available data for herbivorous wild birds, because life in captivity is usually associated with stress or a much poorer and less varied diet, which may translate into animal health. Therefore, in the context of the possible use of the collected data to design probiotic preparations for domestic birds, I think that it would be the right approach. I would also suggest to a greater extent use of data obtained from functional PICRUSt analysis in this discussion. As helpful in this topic, I would recommend the work “The avian gut microbiotacommunityphysiology and function in wild bird” doi: 10.1111/jav.01788that will make easier to find more detailed original reports. 

Author Response

Please find below our responses to your queries and see the attachment with the responses to all reviewers and Editor's queries. Thank you.

Reviewer 1:

Microbial  community  profiling  based  on  16S  ribosomal  rRNA  gene  sequencing has proven to be a reliable and reproducible representation  of  phylogenetic  and  taxonomic  categorization  of  bacteria. However, interpreting metagenomic data is a challenge since, low coverage of well-annotated microbial groups in available gene reference databases which makes it impossible to reliably assign all phylotypes and functional gene identities from environmental microbial sequencing libraries. Much lower number closed OTUs versus open OTUs presented in reviewed manuscript shows it clearly. The choice of the target region, algorithms and thresholds used can also lead to different clustering of OTUs. Thus, the conclusions drawn should be carefully. But all that things are most probably well known to Authors.   

RESPONSE: Thank you, this is indeed an exciting topic and yes, we are aware of it. In fact, the second paragraph of the discussion section covers this topic. Please note that thanks to your comments and the comments from the other reviewers, the manuscript was greatly improved.

Therefore, very important part of the manuscript should be the discussion chapter. In Introduction the Authors declare that their study has: (i) the cognitive goal -  'gut' avian microbiota and (ii) their function. In relation to the first part, generally, in my opinion the molecular, bioinformatics and statistic methods and the conclusions are correct. However, because it is well known, that both diet and individual age (the effect clearly visible in humans) have huge impact on gut microbiota of different animals including birds, I wonder, if authors doing analyzes the data, didn't narrow down these parameters too much referring them only to the abundance of bacteria, which may lead to wrong conclusions in this aspect.

RESPONSE: We agree that both diet and age, either separately or combined, have a huge impact on the composition and function of the gut microbiota. However, in this current study we only collected and presented this as general information about the samples (in other words, this study did not aim to investigate the effect of either factor on the gut microbiota). Moreover, it would be risky to correlate the results with either diet or age because the diet composition did not include nutrient profile (e.g. percentage of protein) and the ages were estimated. Nonetheless, we did use the type of diet in MaAsLin analyses, albeit cautiously, to investigate a potential effect of diet. Regardless of these findings, we used your valuable thoughts and the suggestions of the editor and other reviewers to improve the discussion (see last paragraph of the discussion).

Looking at the data obtained, it seems to be possible, that diet components influence microbiota of considered groups of birds. The group PK3 in comparison to other PK where much younger and on different diet (no legumes included which contain many bioactive compounds with different impact on microorganisms), also both PH4 and PH5 in comparison to other PH differed in age and diet, as well birds in MP4 group were much older than in other MP groups. Moreover, these factors don’t have to equally affect the individual groups of bacteria, but only some, not necessarily the most numerous. Therefore, I would suggest a more detailed and critical discussion on this aspect. In my opinion, it would be good to compare the obtained data with the available data for herbivorous wild birds, because life in captivity is usually associated with stress or a much poorer and less varied diet, which may translate into animal health. Therefore, in the context of the possible use of the collected data to design probiotic preparations for domestic birds, I think that it would be the right approach.

RESPONSE: We very much appreciate your comments and suggestions, and agree with you regarding the possible influence of diet. However, we would also like to be cautious in putting too much weight into this topic because we did not analyze the nutrient profile of each diet (e.g. protein percentage) and did not investigate individual dietary preferences within a mixture of seeds. Nonetheless, we improved the discussion section with a new paragraph discussing possible differences with wild species, and also expanded the discussion on Lactobacillus.

I would also suggest to a greater extent use of data obtained from functional PICRUSt analysis in this discussion. As helpful in this topic, I would recommend the work “The avian gut microbiota: community, physiology and function in wild birds” doi: 10.1111/jav.01788, that will make easier to find more detailed original reports.

RESPONSE: We agree that the results from PICRUSt analysis can be helpful in the discussion but due to the complexity and uncertainties around prediction of functional traits, we decided to expand the Supplementary Information instead. We also found the work published by Grond et al. (2018) particularly useful and cited it in the original and the revised manuscript. However, the methodology employed by the authors to present and compare microbial compositions is difficult to grasp (e.g. Figure 2 shows the core microbiota of mammals, based on a wide range of species presented by Ley et al. 2008; humans, based on a single study that used clone libraries; domestic chickens; and wild birds, comprising a complex mixture of more than 30 bird species). We sincerely hope that you can find the revised manuscript more useful for the scientific community.   

Reviewer 2 Report

The paper entitled "Composition and Potential Function of Fecal Bacterial Microbiota from Six Pet Bird Species" reports the definition of pet Birds fecal microorganisms obtained by 16S rRNA gene analysis.

The paper topic is relevant for the interested scientific community, it is well written and easy-readeble.

Methods and results are clearly and widely described. Discussion and conclusions are conveniently arisen from the obtained results. 

Author Response

Please find below our responses to your queries and see the attachment with the responses to all reviewers and Editor's queries.
Thank you.

Reviewer 2:

The paper entitled "Composition and Potential Function of Fecal Bacterial Microbiota from Six Pet Bird Species" reports the definition of pet Birds fecal microorganisms obtained by 16S rRNA gene analysis. The paper topic is relevant for the interested scientific community, it is well written and easy-readable. Methods and results are clearly and widely described. Discussion and conclusions are conveniently arisen from the obtained results.

RESPONSE: We thank you for your valuable comments.

Reviewer 3 Report

1. Introduction - please justify the chosen method. Present why did you choose 16Sr RNA gene sequencing. Present some references of other similar studies that used the same method. 2./2.1. Please include a briefly presentation of the fecal samples collection methodology, from your previous publication. 2.2. Please clarify if you analysed each fecal sample you collected from each bird, or if you analysed a polled sample. If you mixed all fecal samples you need to describe the method you used. Please specify if you have done some correlations with the birds age. If you did, include it into the manuscript, on the methodology and results. Did you had only one collection/bird or you collect fecal samples periodically? Please highlight if there were differences on the identified microbiome between birds within the species. Detail the maintenance conditions of the birds. Specify if the birds of the same species came from the same breeder. Or a breeder had birds from several species. Some correlations with birds age it would been interesting. To see if the probiotics using in pet birds can be specific to some age group. On Discussions - You presented a product name for a probiotic and the company name. That company funded the research? If don't, please delete the commercial name of the product and the company. Please detail the conclusions with the main microbial species identified and the main correlations from the study.

Author Response

Please find below our responses to your queries and see the attachment with the responses to all reviewers and Editor's queries.
Thank you.

Reviewer 3:

  1. Introduction - please justify the chosen method. Present why did you choose 16Sr RNA gene sequencing. Present some references of other similar studies that used the same method.

RESPONSE: We have modified the introduction to highlight some of the disadvantages of traditional culture techniques and some of the advantages of 16S sequencing used to study microorganisms. We also highlighted the fact that all the studies we cited from birds, including wild species, have used molecular methods.

2./2.1. Please include a briefly presentation of the fecal samples collection methodology, from your previous publication.

RESPONSE: The presentation of our fecal sample collection methodology was included in the original submission in 2.1 (“Briefly, we changed the tray on the bottom of each cage, cleaned it, placed new clean paper on it, and waited for the animals to defecate. This procedure usually took 20-40 minutes until we gathered enough material to fill one 2-mL sterile plastic tube”) but it was improved in the revised manuscript (“Briefly, we changed the tray on the bottom of each cage, cleaned it, placed new clean paper on it, and waited for the animals to defecate. This procedure usually took 20-40 minutes until we gathered enough fecal material to fill one 2-mL sterile plastic tube. The tubes were filled with small aliquots of all droppings available”). Please note that this methodology was not designed to discard potential sources of contamination (e.g. birds walking on feces) but instead to provide a view of the microbiota in feces of pet birds that people have contact with.

2.2. Please clarify if you analysed each fecal sample you collected from each bird, or if you analysed a polled sample. If you mixed all fecal samples you need to describe the method you used.

RESPONSE: The methodology described in 2.1 was improved (see our response to you above) and we clarified in 2.2 that we analyzed 30 pooled fecal samples.

Please specify if you have done some correlations with the birds age. If you did, include it into the manuscript, on the methodology and results.

RESPONSE: Some of the birds’ owners provided us with an estimated age of the birds but this information is not accurate nor precise, which is why we labelled that column in Table 1 as “estimated age”. Moreover, some of the owners did not have information about this at all (labelled “undetermined” in Table 1). Therefore, it is not feasible to correlate the estimated age of the birds with fecal microbiota composition or function. This is now included in the last paragraph of the discussion section. 

Did you had only one collection/bird or you collect fecal samples periodically?

RESPONSE: Unfortunately, we did not collect fecal samples periodically. This was highlighted in 2.1.

Please highlight if there were differences on the identified microbiome between birds within the species.

RESPONSE: This information was presented at the phylum and family level in the original manuscript. We have further highlighted this information in 3.1. 

Detail the maintenance conditions of the birds.

RESPONSE: All birds belonged to different privately-owned flocks that were kept outdoors in small cages with water and food available ad libitum. This information is now included in 2.1.

Specify if the birds of the same species came from the same breeder. Or a breeder had birds from several species.

RESPONSE: All birds came from different owners and this information is included in 2.1 and in Table 1.

Some correlations with birds age it would been interesting. To see if the probiotics using in pet birds can be specific to some age group.

RESPONSE: Some of the birds’ owners provided us with an estimated age of the birds but this information is not accurate nor precise, which is why we labelled that column as “estimated age” in Table 1. Moreover, some of the owners did not have information about this at all (labelled “undetermined” in Table 1). Therefore, it is not feasible to correlate the estimated age of the birds with fecal microbiota composition or function. This is now included in the last paragraph of the discussion section. 

On Discussions - You presented a product name for a probiotic and the company name. That company funded the research? If don't, please delete the commercial name of the product and the company.

RESPONSE: The company did not fund this research. We have removed the commercial name of the product and the name of the company.

Please detail the conclusions with the main microbial species identified and the main correlations from the study.

RESPONSE: The conclusion paragraph was greatly improved based on your valuable suggestions.

Reviewer 4 Report

The manuscript is well written and includes important information on fecal bacteria of pet birds. 

I hope that the manuscript can be improved with the following suggestions.

  1. line 38: In keywords, the authors can select more important words rather than a taxon name, such as Lactobacillus.
  2. line 63-65: I recommend to include the pet species here and address briefly why you chose the species among many others. It can be further explained in the method section. 
  3. line 68-80: As it was indicated in the introduction, diet is one of the most important factors to affect the fecal bacterial communities in birds. Thus, the diet of the sampled pet birds should be addressed here. 
  4. line 68-80: I strongly recommend to address the reason why you pooled samples instead of dealing with individual samples.
  5. line 170, 177: As I understand, each sample indicates a flock pooled with several individuals. In the figure caption, however, it was indicated to be "for each bird species". I think it should be clarified. 
  6. Figure 3 should be modified to avoid an overlap between the legend and the main figure.
  7. In Figure 5, I suggest that authors use "OTUs" instead of "species". Throughout the text, it is often presented that "species (i.e. OTUs at 97% similarity from the open approach)" (in line 270). It is generally true that 97% rRNA similarity can be interpreted as the same species, but different thresholds can be applied (Kim et al. 2014; https://doi.org/10.1099/ijs.0.059774-0) 
  8. line 319-324: I think this paragraph can move to the introduction section or to the end of the discussion section.
  9. line 341-352: Lactobacillus seems to be one of the most important taxa in this study. I expect that more discussions can be described here related to their history and diet. 

Author Response

Please find below our responses to your queries and see the attachment with the responses to all reviewers and Editor's queries.
Thank you.

Reviewer 4:

The manuscript is well written and includes important information on fecal bacteria of pet birds. I hope that the manuscript can be improved with the following suggestions.

RESPONSE: Thank you, we appreciate your valuable suggestions.

line 38: In keywords, the authors can select more important words rather than a taxon name, such as Lactobacillus.

RESPONSE: We replaced Lactobacillus for lactic acid bacteria. 

line 63-65: I recommend to include the pet species here and address briefly why you chose the species among many others. It can be further explained in the method section.

RESPONSE: We chose those species because they are the most common in our area and have included this reason in 2.1. The names of the bird species are included in Table 1.

line 68-80: As it was indicated in the introduction, diet is one of the most important factors to affect the fecal bacterial communities in birds. Thus, the diet of the sampled pet birds should be addressed here.

RESPONSE: We agree that diet is one of the most important factors that affects the fecal microbiota in birds and that is why we included this important information in Table 1. However, please note that our information about diets did not include the nutrient profile, for example protein level, or individual dietary preferences over one particular seed type from a mixture of seeds. Therefore, as you can appreciate, we preferred not to dig much into this, with the exception of MaAsLin analyses.

line 68-80: I strongly recommend to address the reason why you pooled samples instead of dealing with individual samples.

RESPONSE: This is an excellent question. The objective of this study was to describe the microbiota in feces of pet birds that people have contact with, as stated in 2.1. We could indeed have analyzed fecal samples from individual birds but in our experience, birds get anxious when separated from their flock (note that most people have more than 1 bird per cage) and this may impact physiological processes that could modify the consistency of feces which has been shown to be the most influential factor affecting fecal microbiome variation in humans (doi: 10.1126/science.aad3503). This reasoning was included in the last paragraph of the discussion.

line 170, 177: As I understand, each sample indicates a flock pooled with several individuals. In the figure caption, however, it was indicated to be "for each bird species". I think it should be clarified.

RESPONSE: Thanks! We added the word “pooled” in the captions for figure 1 and 2 to clarify.

Figure 3 should be modified to avoid an overlap between the legend and the main figure.

RESPONSE: The parameters in the galaxy platform were modified to avoid the overlap.

In Figure 5, I suggest that authors use "OTUs" instead of "species". Throughout the text, it is often presented that "species (i.e. OTUs at 97% similarity from the open approach)" (in line 270). It is generally true that 97% rRNA similarity can be interpreted as the same species, but different thresholds can be applied (Kim et al. 2014; https://doi.org/10.1099/ijs.0.059774-0).

RESPONSE: We definitely agree with you and replaced “species” for “OTUs” when needed.

line 319-324: I think this paragraph can move to the introduction section or to the end of the discussion section.

RESPONSE: The aim of this small paragraph is to provide a brief summary to introduce the reader into the discussion. We rephrased this paragraph to better reflect this.

line 341-352: Lactobacillus seems to be one of the most important taxa in this study. I expect that more discussions can be described here related to their history and diet.

RESPONSE: We agree with the reviewer and that is why in the original manuscript we devoted one paragraph to the discussion about Lactobacillus. Thanks to your suggestions, this paragraph was expanded to better reflect the possible relevance of Lactobacillus in the avian microbiota.

Round 2

Reviewer 1 Report

As all my suggestions were addressed in the revised version (birds-1036367-peer-review-v2) of the manuscript I can recommend the paper for publication in Birds. 

Author Response

On behalf of all my coauthors, I would like to thank you for your valuable time and suggestions.

Sincerely,

Jose F. Garcia-Mazcorro